# A System-Independent Derivation of Preferential Attachment from the Principle of Least Effort

**DOI:** 10.3390/e25020305

**Published:** 2023-02-07

**Authors:** François Xavier Machu, Ru Julie Wang, Jean Louis Cheng, Jeremy Cocks, Qiuping Alexandre Wang

**Affiliations:** 1Laboratoire SCIQ, ESIEA Group, 9 Rue Vésale, 75005 Paris, France; 2IMMM, CNRS UMR 6283, Le Mans Université, Ave. O. Messiaen, 72085 Le Mans, France

**Keywords:** preferential attachment, least effort, maximum efficiency, power law, calculus of variation

## Abstract

Preferential attachment (PA) is a widely observed behavior in many living systems and has been used in modeling many networks. The aim of this work is to show that the mechanism of PA is a consequence of the fundamental principle of least effort. We derive PA directly from this principle in maximizing an efficiency function. This approach not only allows a better understanding of the different PA mechanisms already reported but also naturally extends these mechanisms with a non-power law probability of attachment. The possibility of using the efficiency function as a general measure of attachment efficiency is also investigated.

## 1. Introduction

Preferential attachment (PA) is a widely observed mechanism of evolution for systems or networks in a state of growth in which certain quantities, such as wealth, information, connections, etc., are distributed among the agents of the systems according to how much of these quantities they already possess [1,2,3,4]. For example, in a growing network, PA implies that the more a node is connected to other nodes, the more likely it is to be linked to new nodes; in the process of searching for information on the internet, more connected pages tend to attract more connections; in relationship networks, a person who has more friends tends to attract more new friends than a person who has fewer friends, and so on and so forth. This preferential effect is well-summarized by statements such as “the rich are getting richer”, “money makes money” or “attraction attracts”.

One of the motivations behind the introduction of PA as a model of some complex systems is its potential to yield power law or near-power law distributions [1,2,3,4,5,6]. These kinds of distribution laws are widely observed in nature, especially in living systems or systems driven by living agents, such as natural languages, social networks, economic systems, information networks, biological systems and so forth [1,2,3,7,8,9]. The first application of PA dates back to Yule in his study of species evolution [1], followed by the work of Simon [2] on city size and of Price [3] on the citation of documents. PA became more popular after the work of Barabasi and Albert on the construction of scale-free networks [4]. More recent work [5,6] has shown that networks modeled with linear PA have power law distribution. An example of linear PA is the probability p(k) for a node of degree *k* in a growing network being connected with a new node is proportional to *k*: p(k)∝k. However, when PA is nonlinear with a power probability of connection $p\left(k\right)\propto k^{\gamma }$, where the power γ is a positive real number, which can be different from one, the system does not necessarily have a power law distribution [5].

Nowadays, PA has become a powerful mechanism in modeling work, capable of accounting for power law or near-power law distributions of a larger number of networks with different natures and properties [10]. The following questions can be asked about why these linear and nonlinear preferential mechanisms should be considered: whether more complicated functional forms of p(k) can be useful, and what these more complicated forms are. There are already efforts to use different forms of p(k) to generate scale-free network, or to choose p(k) using the maximum likelihood method [11,12,13]. A more fundamental question is whether there are underlying principles allowing a deeper understanding of the PA mechanism and a clearer vision of its potential application to more systems.

The aim of this work is to address the last question. The main contribution is the derivation of models of PA directly from an appealing fundamental principle called the principle of least effort [14]. According to this principle, most, if not all, living systems must obey this rule in order to perform something in the most economical way, even for survival. This principle was first proposed by Ferrero in a study of human behavior [7]. It was later extended to other living systems by Zipf in a quantitative study of the distribution laws of linguistic and social systems: *everything carried out by human beings and other biological entities must be undertaken with the least effort (at least statistically)* [7]. In what follows, we will refer to this principle as the Ferrero–Zipf principle (FZP). The FZP is very well summarized in the short proverb “achieving more by doing less”, a statement highlighting two aspects of the FZP: maximum achievement and minimum effort. We believe these two aspects should be considered the essence of the FZP.

Recently, the FZP has been successfully used in a derivation of power laws with a system-independent calculus of variation [15,16]; we will provide more details about this below. This previous work motivated us to go further in the application of the FZP. The present work is a step forward in applying the FZP to derive a PA mechanism. We would like to stress that the behavior of PA is observed in most, if not all, living systems or in systems driven by living agents (internet, languages, social and informational networks, etc.), where many of these systems are capable of making an effort to achieve something. From this point of view, the application of the FZP to these systems is relevant and legitimate, especially when it concerns the activities or properties relating to what they spend and achieve and the efficiency of their activities. Questions may arise about the legitimacy of the FZP for living systems that are incapable of making efforts, for example, many plant-based systems or protein networks. This question will be addressed below in the discussion of an equivalent principle called the principle of maximum efficiency.

We would also like to stress that deriving the PA models from fundamental principles does not use the principles to model scale-free systems nor modify PA models to yield power laws, as was undertaken in [17]. For us, to be able to derive a model using a fundamental principle is both a confirmation of the validity and usefulness of the principle itself, as well as a confirmation of the robustness of the model. If the model is derivable from a widely-accepted fundamental principle, then the veracity and the applicability of the model are reinforced.

## 2. Least Effort and Maximum Efficiency

If we want to implement the FZP through maximum/minimum calculus, we must both maximize the achievement and minimize the effort. This idea led us to a mathematical implementation of the FZP, i.e., maximizing efficiency (defined as the ratio of achievement *X* over effort *Q*) [15]. We referred to this approach in our previous paper as the principle of maximum efficiency (MAXEFF) [15]. MAXEFF is not only easier to implement mathematically than the original FZP, as discussed in [15], but can also possibly be applied to living systems that show scale-free properties even if these systems do not “know” how to make an effort, such as plant-based biological systems and protein networks [17,18]. In other words, a system incapable of making efforts can perfectly maximize its dynamic efficiency or minimize its costs [19]. Hence, MAXEFF is not only easier to apply than the FZP with a clear definition of efficiency but is also more general and useful than the FZF in the sense that it can be used with more living systems.

MAXEFF has recently been applied to the derivation of the Zipf–Pareto laws [7,8,9] using a functional of efficiency η for living systems composed of a large number of living agents [15,16,20]. This efficiency as a functional of the continuous probability distribution ρ(x) is defined as follows:(1)$$\eta =\int_{x_{min}}^{x_{max}}\frac{\rho {\left(x\right)}^{1-a}-1}{a}dx$$
in which *a* is a real number, ρ(x) is the probability density distribution for an agent to be at the value x of an observable X. X can be any achievement (money, information, frequency of words, population, friends, number of connections, etc.) that living agents are trying to obtain through effort (*Q*). Note that *Q* can also represent quantities of the same nature as *X* (money, information, time, and so on). The more X the agents achieve, the larger is the value x of *X*. The efficiency of an agent *i* is defined by e=X/Q. This efficiency should be a function of the probability density distribution ρ(x) for finding an agent whose achievement is x. This leads to the average $\eta =\int_{x_{min}}^{x_{max}}e\left[\rho \left(x\right)\right]\rho \left(x\right)dx$ [15,16]. The system-independent efficiency function in Equation (1) was derived from an extension (see Equation (2) below) of the nonadditive relationship $e=e_{1}+e_{2}-e_{1}e_{2}$ of thermodynamic engines [15], where $e_{1}$ and $e_{2}$ are the efficiency of two independent engines (1 and 2, respectively) and e is the efficiency of the engine composed of engines 1 and 2 connected in series [16]. The nonadditive relation above is extended to living agents (or nodes of growing networks) as follows:(2)$$e=e_{1}+e_{2}+ae_{1}e_{2}\;$$
where $e_{1}$ and $e_{2}$ are the efficiency of two living agents regarded as engines 1 and 2, and e is the efficiency of the composite living agent composed of agents 1 and 2. The parameter *a* characterizes the efficiency e=−X/aQ for non-thermodynamic systems. In growing networks, achievement *X* represents the number of connections (degree) of a node; the effort *Q* is related to the cost of the connections (an example of cost estimation can be seen in [17]); and *e* is the connection efficiency between nodes. The parameter *a* characterizes the relationship between *X* and *Q*. For thermodynamic engines, a=−1 because the thermodynamic efficiency given by e=X/Q must be positive and smaller than unity due to the energy conservation relationship between the output work *X* and the input heat *Q*. However, the quantities *X* and *Q* involved in the processes in living systems are not necessarily bound by the law of energy conservation. There is no necessarily conservative relationship between *X* and *Q* (both can be money, information, time, number of connections, population, frequency of words, and so forth). So a can be different from −1, leading to efficiency being larger than one and sometimes even being negative [16]. We have shown if the efficiency of a system satisfies Equation (2), then Equation (1) is the only expression of the efficiency of that system as a function of the probability distribution ρ(x). A proof of this uniqueness of the efficiency function is formulated in [16], and the complete proof is given in [20].

In what follows, we will show that the linear and nonlinear PA mechanisms are all direct consequences of the FZP implemented by MAXEFF using the efficiency function in Equation (1). We also show that this efficiency formula provides a possible measure of the average performance of agents in attachment processes.

## 3. From Least Effort to Preferential Attachment

Suppose a system is composed of a large number of agents, all trying to achieve a quantity represented by the variable *X*. In the context of system evolution through new connections being made between agents, achievement *X* can be anything that an agent obtains by attaching itself to other agents. From this point of view, *X* is the quality of agents attracting the other agents. In this case, ρ(x) should be interpreted as the probability density for an agent being connected to another agent with a value x of *X*. *X* can be information, money, number of connections (degrees), population, frequency of events, group size, resources in biological systems, and so on.

The average efficiency of the quest by all the agents is given by Equation (1) and can be maximized by introducing the constraint of constant average achievement. It is worth mentioning that the systems considered are still undergoing growth or evolution; hence, this maximization does not mean maximum efficiency in time. Instead, this maximization provides the maximum efficiency corresponding to the optimal stable distribution ρ(x) at a stationary stage of evolution. In other words, the efficiency will be lower for any possible distributions other than the optimal one. The constraint of constant average achievement (the degrees in growing networks) implies that this average quantity must remain constant for different possible distributions at that stage of evolution. Suppose the average of the achievement of all the agents is provided by $\bar{X}=\int_{x_{min}}^{x_{max}}x\rho \left(x\right)dx$; the functional to be maximized should be the sum ($\eta +c\bar{X}$), where c is the multiplier of the constraint $\bar{X}$. The variation is provided by $\delta \left(\eta +c\bar{X}\right)=0$, which straightforwardly leads to a power law:(3)$$\rho \left(x\right)=Cx^{\gamma }$$
where $\gamma =-\frac{1}{a}$ and $C=1/\int_{x_{min}}^{x_{max}}x^{\gamma }dx$ is the constant of normalization [15]. When a<0 and γ > 0, ρ(x) provides the connection probability of the usual PA models [4,5,6]. In the case of discrete values $x_{i}$ of X (degree or number of connections for instance), the Equation (1) should be expressed as a function of the discrete probability $p_{i}$ of $x_{i}$: $\eta =\frac{\sum p_{i}^{1-a}-1}{a}$. The above calculus will lead to the connection probability $p_{i}=Cx_{i}^{\gamma }$ for an agent to be connected to agents with the degree $x_{i}$. The Barabasi–Albert model of linear PA corresponds to γ=1, leading to scale-free networks (power-law degree distribution) [4]. γ < 1 is the sublinear PA that leads to a stretched exponential degree distribution [5], and γ > 1 is the superlinear PA that leads to star graph networks (one central node connected to all other nodes, each having only one connection to the central node) and other more complicated degree distributions depend on the range of γ [6].

The above calculus of variation can be extended by adding the normalization $\int_{x_{min}}^{x_{max}}\rho \left(x\right)dx=1$ as a second constraint. This normalization was already used above in Equation (3) but it is not a constraint of the variational calculus. Now, it will be used as a constraint with the multiplier *c*’ in order to introduce one more parameter into the probability distribution, implying the calculus:(4)$$\delta \left(\eta +c\bar{X}+c^{\prime }\int_{x_{min}}^{x_{max}}\rho \left(x\right)dx\right)=0$$
This is equivalent to $\frac{\partial }{\partial \rho }\left(\frac{\rho^{1-a}-1}{a}+c\rho x+c^{\prime }\rho \right)=\frac{1-a}{a}\rho^{-a}+cx+c^{\prime }=0$, leading to:(5)$$\rho \left(x\right)=C{\left(x+\beta \right)}^{\gamma }$$
where $\beta =\frac{c^{\prime }}{c}$, $\gamma =-\frac{1}{a}$ and $C=1/\int_{x_{min}}^{x_{max}}{\left(x+\beta \right)}^{\gamma }dx$ is the normalization constant. In the discrete case, we have $p_{i}=C{\left(x_{i}+\beta \right)}^{\gamma }$ and $C=1/\sum_{i}^{w}{\left(x_{i}+\beta \right)}^{\gamma }$, where the summation is over all the w possible values $x_{i}$ of X. Equation (5) can be called the extended PA mechanism (ePA) with one more parameter than in Equation (4). When β=0, ePA becomes the usual PA. The role of β obviously increases the probability of connection for nodes that have x < β. An application of this ePA probability of connection has recently been reported [12] as a parametric model of PA, including the sublinear model with γ<1, and an affine linear model with γ=1, leading to a power-law degree distribution with a variable exponent.

## 4. Attachment Efficiency

Equation (1), as a measure of efficiency, was investigated in our previous work as a function of the parameters of the Zipf–Pareto distribution laws [15,16]. One of the advantages of this measure is that it can be estimated for any scale-free (power law) system whenever the distribution law becomes known after either empirical study or mathematical analysis. The result shows that efficiency increases as the Pareto law exponent decreases, flattening the curve of the distribution. For example, a higher number of people have higher incomes, and this decrease in the Pareto exponent brings increasing inequality, as well as greater probabilistic uncertainty in the distribution [16]. This efficiency behavior is expected with Pareto distributions and hints at the usefulness of Equation (1) as a reliable system-independent efficiency measure.

The efficiency of preferential attachment in the construction of network can be evaluated by using Equation (1) which gives, for the distribution Equation (5) of the ePA model
$$\eta =\int_{0}^{r}\frac{\rho {\left(x\right)}^{1-a}-1}{a}dx=\int_{0}^{r}\frac{C^{1+\frac{1}{\gamma }}{\left(x+\beta \right)}^{\gamma +1}-1}{a}dx$$
with the following result:(6)$$\eta =\gamma r-{\left[\frac{\gamma \left(\gamma +1\right)}{{\left(r+\beta \right)}^{\gamma +1}-\beta^{\gamma +1}}\right]}^{1+\frac{1}{\gamma }}\frac{{\left(r+\beta \right)}^{2+\gamma }-\beta^{\gamma +2}}{2+\gamma }$$
where $C=\frac{\gamma +1}{{\left(r+\beta \right)}^{\gamma +1}-\beta^{\gamma +1}}$ is the normalization constant in Equation (5) and r is the maximum value of x. If β=0, we yield $\eta =r\gamma -\gamma \left(\frac{{\left(\gamma +1\right)}^{1+\frac{1}{\gamma }}r^{-\frac{1}{\gamma }}}{2+\gamma }\right)$. In general, it can be verified that $C<{\left(\frac{\beta^{\left(\gamma +2\right)}+r\left(\gamma +2\right)}{{\left(r+\beta \right)}^{2+\gamma }}\right)}^{\frac{1}{1+\frac{1}{\gamma }}}$, which implies η>0. The variation of efficiency in Equation (6) as a function of parameters γ, β and r is illustrated in Figure 1 and Figure 2. We have noticed that, although β has an important influence on the ePA distribution p(x) of Equation (5), efficiency depends little on β, which is a consequence of Equation (6), where similar β-dependent factors are found in both the denominator and numerator.

An essential common characteristic in the above figures is that efficiency increases with increasing γ and r, which is expected because the increase of γ implies an increasing likelihood of attachment of new nodes to the nodes of larger values of *x*. The increase of r, the largest value of the achievement *x* (the degree or the number of connections), implies increasing contributions from a larger and larger number of connections. Hence, increasing γ and r necessarily increases the average value of the achievement *X* (as shown in Figure 3) and, as a consequence, increases the efficiency of the process of connection. The average value of *X* can be calculated as follows:(7)$$E\left(X\right)=\int_{0}^{r}x\;\rho \left(x\right)dx=\int_{0}^{r}Cx{\left(x+\beta \right)}^{\gamma }dx$$
The result is $E\left(X\right)=\frac{1}{{\left(r+\beta \right)}^{\gamma +1}-\beta^{\gamma +1}}\left[r{\left(r+\beta \right)}^{\gamma +1}-\frac{1}{\left(\gamma +2\right)}\left(r^{\gamma +2}-\beta^{\gamma +2}\right)\right]$. The variation of E(X) as a function of two parameters, γ and r (for β=0), is plotted in Figure 3, where we see an increasing average of the achieved *X* with increasing γ and r.

In order to show the pertinence of the formula in Equation (1) as a measure of efficiency, and to have an idea about the γ and r dependences of the efficiency in real networks, we have found empirical data from several real networks reported in [19] and calculated their efficiency using Equation (1). In fact, the data from [19] are all about scale-free networks with power-law degree distributions $p\left(x\right)\propto x^{\gamma^{\prime }}$ with values of $\gamma^{\prime }$ ranging from 1 to 3.5, meaning that their efficiency should be calculated with γ=1 for all the networks according to the common view that linear PA yields power laws [1,2,3,4,5,6,12]. The maximum degree *r* of these networks varies in the interval $10<r<10^{8}$. The results are plotted in Figure 4 for nine networks. The abscissa is the exponent $\gamma^{\prime }$ of the degree distribution in order to separate the different networks. We see that the efficiency η of these networks is mainly determined by r because of its very large variation from one network to another. As for the γ dependence of the efficiency, we can imagine with Figure 1 and Figure 2, that γ varying in the vicinity of 1 [5,6,12] would have little impact on the variation of the efficiency.

It is noteworthy that the scientific citation network has much higher values of r and η than the other networks, whose efficiencies are all similar because of the similar values of *r*. This phenomenon can be understood as a consequence of the attraction of scientific journals and papers promoted by their citation indices (impact factor, h-index, etc.). This information is publicly accessible to researchers. The same promotional mechanism can also work for telephone call networks and communication networks, where enhancing the preferential effect can increase the efficiency of preferential attachment.

## 5. Concluding Remarks

We have shown that the important mechanism of preferential attachment, which has been widely observed in the evolution of many living systems, can be generated from the fundamental principle of least effort with the help of a universal (system-independent) efficiency function. It has been demonstrated that this approach can generate not only the usual linear and nonlinear PA models but also a more general form of connection probability, which has proven useful in parametric models [12]. We have also shown that this functional can be used as a general measure of the efficiency of attachment for many types of systems. Further empirical study and numerical simulation of evolutionary processes are necessary to confirm the usefulness and accuracy of this general measure of efficiency.

The fact that the PA mechanisms have been derived from the fundamental principles, the FZP and MAXEFF, provides us with a deeper understanding of PA mechanisms and more confidence in applying them to the modeling of other systems in which FZP or MAXEFF is at play. On the other hand, this derivation of PA also tells us that FZP is not necessarily associated with a power law distribution, as many believe; as mentioned in the introduction, nonlinear PA models with γ≠1 have yielded non-power law distributions [5,12].

Our last remark is that the derivation of PA from the FZP and MAXEFF does not mean that these principles are the only origin of PA behavior. Other system-specific mechanisms may be at play. However, it is worth noting that, due to the fundamentality and the universality of the FZP, using it may be a simpler and more reliable approach than using different system-specific mechanisms to thoroughly understand PA behaviors as a common property widely observed in a large number of living systems with very different natures, characteristics and properties. Is it not just the least effort and more efficient approach?

## Figures and Tables

**Figure 1 entropy-25-00305-f001:**
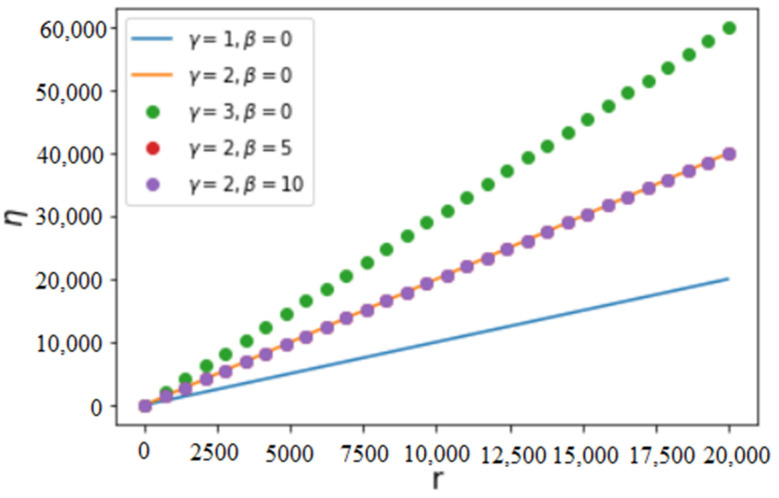
Variation of efficiency η of attachment as a function of *r* for different fixed values of β and γ. We notice that η increases almost linearly with increasing *r* and depends little on β.

**Figure 2 entropy-25-00305-f002:**
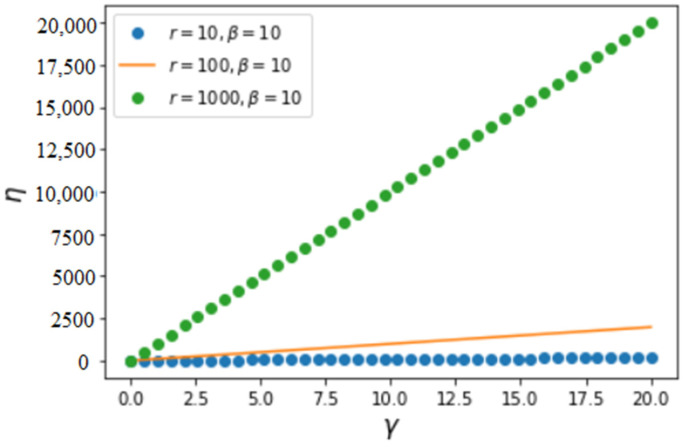
Variation of efficiency η of attachment as a function of γ with fixed β=10 and several values of r. We notice that η increases almost linearly with increasing γ in this interval of variation.

**Figure 3 entropy-25-00305-f003:**
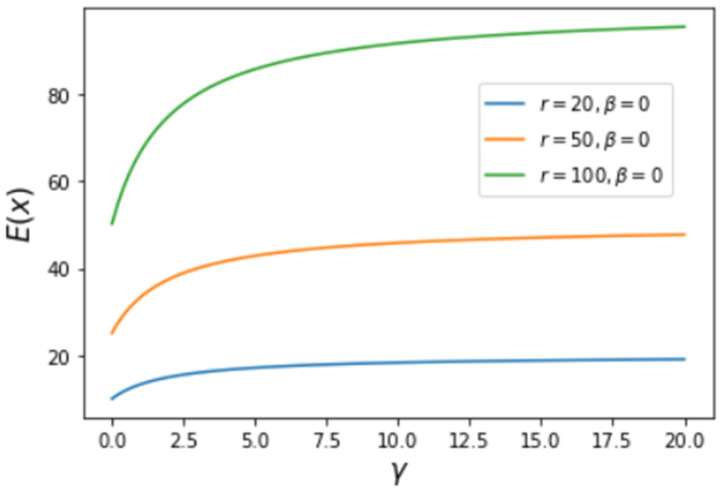
Variation of the average of *x* as a function of γ with several values of β and r.

**Figure 4 entropy-25-00305-f004:**
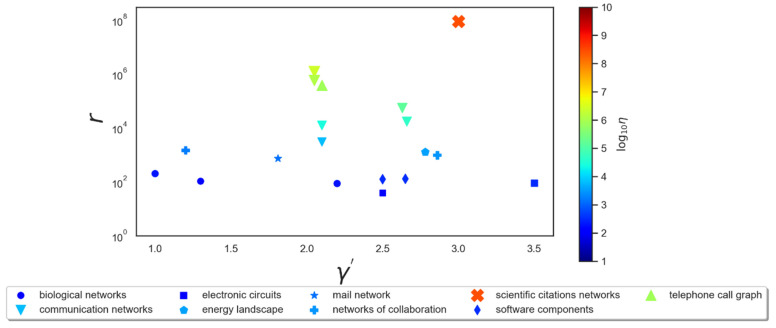
Illustration of certain values of efficiency *η* (vertical axis right) calculated from empirical values of r found in [20] for different networks. We chose *γ* = 1 because all these networks are described with the degree distribution $p\left(x\right)\propto x^{\gamma^{\prime }}$. The exponent $\gamma^{\prime }$ is represented on the abscissa. The color of the symbols changes from blue to red with increasing value of η.

## Data Availability

Data sharing not applicable.

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
