# Peer review of "A System-Independent Derivation of Preferential Attachment from the Principle of Least Effort"

_entropy, 2023, doi:10.3390/e25020305_

Round 1
Reviewer 1 Report
The manuscript proposes a principle of least effort as an underlying mechanism for preferential attachment. There are two problems with the manuscript.
Firstly, the authors did not discuss existing seminal works. Researchers have considered preferential attachment as an emergent property from some kinds of optimizations many years ago. See Souza et al., "Emergence of tempered preferential attachment from optimization", PNAS (2007). The manuscript did not even cite this paper. With no discussions and comparisons with such important works, the significance of the manuscript is in serious doubt.
Secondly, there is no validation of the proposed model in real-world datasets. Network scientists have proposed countless toy models for complex networks. However, most of them do not fit real-world networks well. Why bother with another one if you do not demonstrate its validity in real-world data?
I do not believe that the authors can fix the above problems in a reasonable amount of time. Therefore, I suggest rejecting the paper.
Reviewer 2 Report
The authors derive the preferential attachment (PA) in growing systems from the principle of least effort proposed in the authors' previous work. How the efficiency of PA is dependent on the parameters is investigated. Further, the efficiency of PA in empirical networks is estimated. It is shown that the efficiency of PA is mainly dependent on the value of the maximum degree and not so sensitive to the exponent of the power law of PA.
The approach taken in this paper seems to make sense and is interesting. However, the reviewer thinks that much improvement is needed to publish the paper. The following issues should be considered.
Major issues:
1.
Interpretation of MAXEFF in the context of the growing network models should be discussed more thoroughly. Without this, Sec.3 is almost a repetition of the authors' previous work. In particular, explain the meanings of the followings in the growing network models:
(i) Equation (2),
(ii) The constraint for the variational problem leading to Eq.(3),
(iii) The second constraint for the variational problem leading to Eq.(4) (corresponding to the multiplier c').
These considerations will also clarify the limitation of the approach. For example, (ii) implies that the average degree is constant as networks grow. Hence, the phenomenon of accelerated growth is out of the scope under the constraint of (ii).
2.
How the exponent of the power-law of PA (gamma) for each empirical network shown in Fig.5 is calculated? In Ref.[19], characteristic quantities for 37 empirical networks are summarized in Table 3.7. Although the exponents of the power-law degree distributions are given (denoted by gamma, which is different from gamma here), the reviewer cannot find the exponents of the power-law of PA there.
3.
At the beginning of Sec.5, the authors write ``We have shown that the important mechanism of preferential attachment, which is widely observed in the evolution of living systems, is nothing but a result of the fundamental principle of least effort.'' The reviewer agrees that the mathematical form of PA can be derived from the principle of least effort. However, this does not necessarily imply that the actual mechanism producing PA in empirical networks is the principle of least effort. To validate the authors' claim, they should explain why the principle of least effort is a more appropriate mechanism than the known mechanisms that give rise to PA such as link selection, duplication and divergence, and so on.
Minor issues:
4. Page 3, Equation (1) (and Page 7, Line 161):
The formula of the efficiency seems to be different from that in Ref.[16]. Is there any reason for the difference?
5. Page 6, Figure 1:
Figure 1 can be deleted because the role of beta in Eq.(5) is evident from Eq.(5) itself.
6. Page 7, Line 167:
``Fig.(1)'' should be ``Fig.1''.
7. Page 8. Line 187:
`C' should be deleted.
Reviewer 3 Report
Usually, in Physics or in other fields of study, reducing a model to a problem of least action or to a principle of least effort reveals a new layer of the model, which is more fundamental in terms of first principles. So the present work contributes to find a more fundamental principle for power law behaviors present in a diversity of systems, and also generalizing this behavior for some nonlinear relations.
So, my recommendation is for the publication of the paper.
One observation, thought: plese correct the use of parenthesis (or better still, brackets) in the equation of lines 187 and 188. Also, this equation should be numbered.
English corrections or suggestions (at the author's discretion). Please change
"if we wants" to "if one wants" or "if we want".
"his important" to "has important"
Round 2
Reviewer 2 Report
It seems that the authors considered the reviewers' comments and revised the manuscript appropriately. The reviewer recommends accepting the manuscript.
Reviewer 3 Report
All issues have been properly addressed by the authros.